# Molecular dissection of box jellyfish venom cytotoxicity highlights an effective venom antidote

Man-Tat Lau[1,2], John Manion [1], Jamie B. Littleboy[1], Lisa Oyston[1], Thang M. Khuong[1], Qiao-Ping Wang[1,3], David T. Nguyen[4], Daniel Hesselson [4,5], Jamie E. Seymour [6] & G. Gregory Neely[1,2]

The box jellyfish *Chironex fleckeri* is extremely venomous, and envenoming causes tissue necrosis, extreme pain and death within minutes after severe exposure. Despite rapid and potent venom action, basic mechanistic insight is lacking. Here we perform molecular dissection of a jellyfish venom-induced cell death pathway by screening for host components required for venom exposure-induced cell death using genome-scale lenti-CRISPR mutagenesis. We identify the peripheral membrane protein ATP2B1, a calcium transporting ATPase, as one host factor required for venom cytotoxicity. Targeting ATP2B1 prevents venom action and confers long lasting protection. Informatics analysis of host genes required for venom cytotoxicity reveal pathways not previously implicated in cell death. We also discover a venom antidote that functions up to 15 minutes after exposure and suppresses tissue necrosis and pain in mice. These results highlight the power of whole genome CRISPR screening to investigate venom mechanisms of action and to rapidly identify new medicines.

[1] The Dr. John and Anne Chong Lab for Functional Genomics, Charles Perkins Centre and School of Life & Environmental Sciences, The University of Sydney, Sydney, NSW 2006, Australia. [2] Genome Editing Initiative, The University of Sydney, Sydney, NSW 2006, Australia. [3] School of Pharmaceutical Sciences (Shenzhen), Sun Yat-sen University, Guangzhou 510275, China. [4] Diabetes and Metabolism Division, Garvan Institute of Medical Research, Darlinghurst, NSW 2010, Australia. [5] St Vincent's Clinical School, UNSW Sydney, Darlinghurst, NSW 2052, Australia. [6] Australian Institute of Tropical Health and Medicine (AITHM) and Centre for Biodiscovery and Molecular Development of Therapeutics (CBMDT), James Cook University, McGregor Road, Smithfield, Cairns, QLD 4878, Australia. Correspondence and requests for materials should be addressed to G.G.N. (email: greg.neely@sydney.edu.au)

The box jellyfish, *Chironex fleckeri*, is one of the most venomous animal in the world[1]. Contact with jellyfish tentacles will trigger the explosive release of nematocysts that deliver potent and rapid-acting venom into the victim or prey. *C. fleckeri* envenoming can be life-threatening, however for the vast majority of cases patients suffer extreme pain and local tissue destruction[2,3]. The venoms of box jellyfish are mixtures of bioactive proteins that can cause potent haemolytic activity, cytotoxicity, membrane pore formation, inflammation, in vivo cardiovascular collapse and lethal effects in experimental animals[4–6]. Importantly, the molecular mechanisms involved in these effects are largely unknown, and here we perform the first genomic characterisation of the venom death pathway.

The classic treatment response for box jellyfish envenoming is to administer an antivenom generated in sheep[7], although the efficacy of this antivenom remains in question[8,9]. More recently, some venom activities have been reported to be suppressed by intravenous zinc[10], or by heating the site of the sting[11]. However, there are currently no therapies that directly target pain and local tissue necrosis, the most common clinical features of envenoming. The major obstacle to developing new therapies is the limited molecular understanding of venom action, a prerequisite for more rational therapies[10].

Recently, the bacterial clustered regularly interspaced short palindromic repeats (CRISPR)-Cas9 system has been shown effective for genome-scale loss of function screens in mammalian cells[12,13]. This approach is particularly suited to identify genes required for drugs or toxins to trigger cell death, and has been used to characterise cell death in response to cancer drugs[12,13],

bacteria toxins[14] and viral infection[15]. To better understand the biology of *C. fleckeri* venom mechanism of action, we perform the genome-scale functional interrogation of box jellyfish venom cytotoxicity, identifying hundreds of host candidate genes and pathways critical for venom action. Moreover, our molecular insights directly informed a rational drug repurposing strategy that identified a new box jellyfish venom antidote that can suppress tissue destruction and attenuate the excruciating pain associated with envenoming.

## Results

**C. fleckeri venom kills cells via necroptosis and apoptosis.** *C. fleckeri* has tentacles up to 3 m long which contain a venom that causes excruciating pain and local tissue damage (Fig. 1a). We found that venom isolated from *C. fleckeri* rapidly killed human cells in a concentration-dependent manner by resazurin-based cell viability assay (Fig. 1b) and similar results were obtained by evaluating LDH release or ATP depletion (Supplementary Fig. 1a, b). To determine the mode of cell death triggered by *C. fleckeri* venom, we pharmacologically blocked apoptotic (Ac-DEVD-CHO) and/or necroptosis pathways (necrosulfonamide; NSA), then treated cells with venom. Venom cytotoxicity was insensitive to caspase inhibition, whereas blocking necrosis with NSA significantly reduced cell death (Fig. 1c). Of note, inhibition of caspase activity with OVD-OPh or Z-VAD-FMK also had no effect on cell death (Supplementary Fig. 1c, d). Moreover, depletion of major pro-apoptotic components (such as BAK1, BAX, BID[16], BOK[17] and CASP8), or the

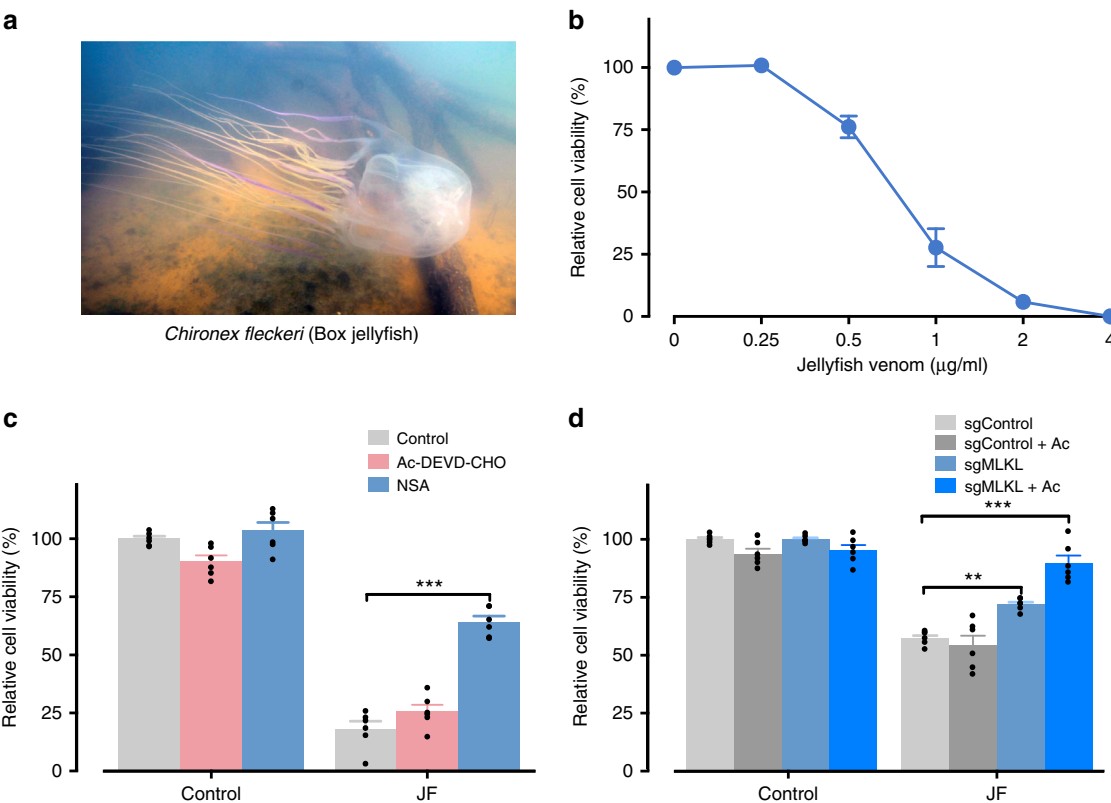

**Fig. 1** Box jellyfish venom induces a predominantly necrotic cell death. **a** Mature *C. fleckeri* jellyfish. **b** HAP1 cells were treated with vehicle or different concentrations of jellyfish venom (as indicated) for 24 h, and cell viability was then determined by resazurin-based cell viability assay ($n = 3$). **c** Inhibition of MLKL by necrosulfonamide (NSA; 5 μM), but not caspase inhibition by Ac-DEVD-CHO (10 μM), reduces the jellyfish venom (1 μg/ml) induced cell death ($n = 6$). **d** Depletion of MLKL conferring resistance to jellyfish venom in HAP1 cells. Inhibition of caspase by Ac-DEVD-CHO (Ac) in MLKL-depletion (sgMLKL) cells protects the jellyfish venom (0.75 μg/ml) induced cell death in a synergistic manner ($n = 6$). All data represented as mean ± S.E.M. one-way ANOVA followed by Tukey's post hoc test, **$p < 0.01$; ***$p < 0.001$

pyroptotic mediator GSDMD[18], did not protect cells from venom cytotoxicity (Supplementary Fig. 1e). Finally, CRISPR targeting of MLKL, a critical mediator of necroptosis[19], provided significant protection from venom cytotoxicity, while additional pharmacological inhibition of caspase had a synergistic protective effect (Fig. 1d). Taken together, these data suggest that box jellyfish venom cytotoxicity involves necroptotic and apoptotic machinery.

**A whole-genome CRISPR screen for box jellyfish venom.** To further investigate the mechanisms of *C. fleckeri* venom cytotoxicity, we performed genome-scale CRISPR knockout (GeCKO) screen. We mutagenised the HAP1 cells with the GeCKO v2 library, which targets 19,050 human genes with 123,411 unique

guide sgRNA sequences[20], and then selected these knockout pools with a lethal concentration of venom for 14 days. We recovered the surviving cells and quantified sgRNA abundance in the selected cells versus an unselected control population by sequencing (Fig. 2a). We observed enrichment of multiple guide RNAs associated with venom resistance (Fig. 2b, c; Supplementary Data 1) and identified a set of the jellyfish venom host factors that were significantly enriched in venom selected cells compared with unselected controls (Fig. 1d, e; Supplementary Data 2).

**ATP2B1 is required for the venom-induced cell death.** We identified *ATP2B1* (ATPase plasma membrane $Ca^{2+}$ transporting 1; also known as *PMCA1*) as one of the top-ranking hits within our screens and the top-ranking membrane protein (Fig. 3a;

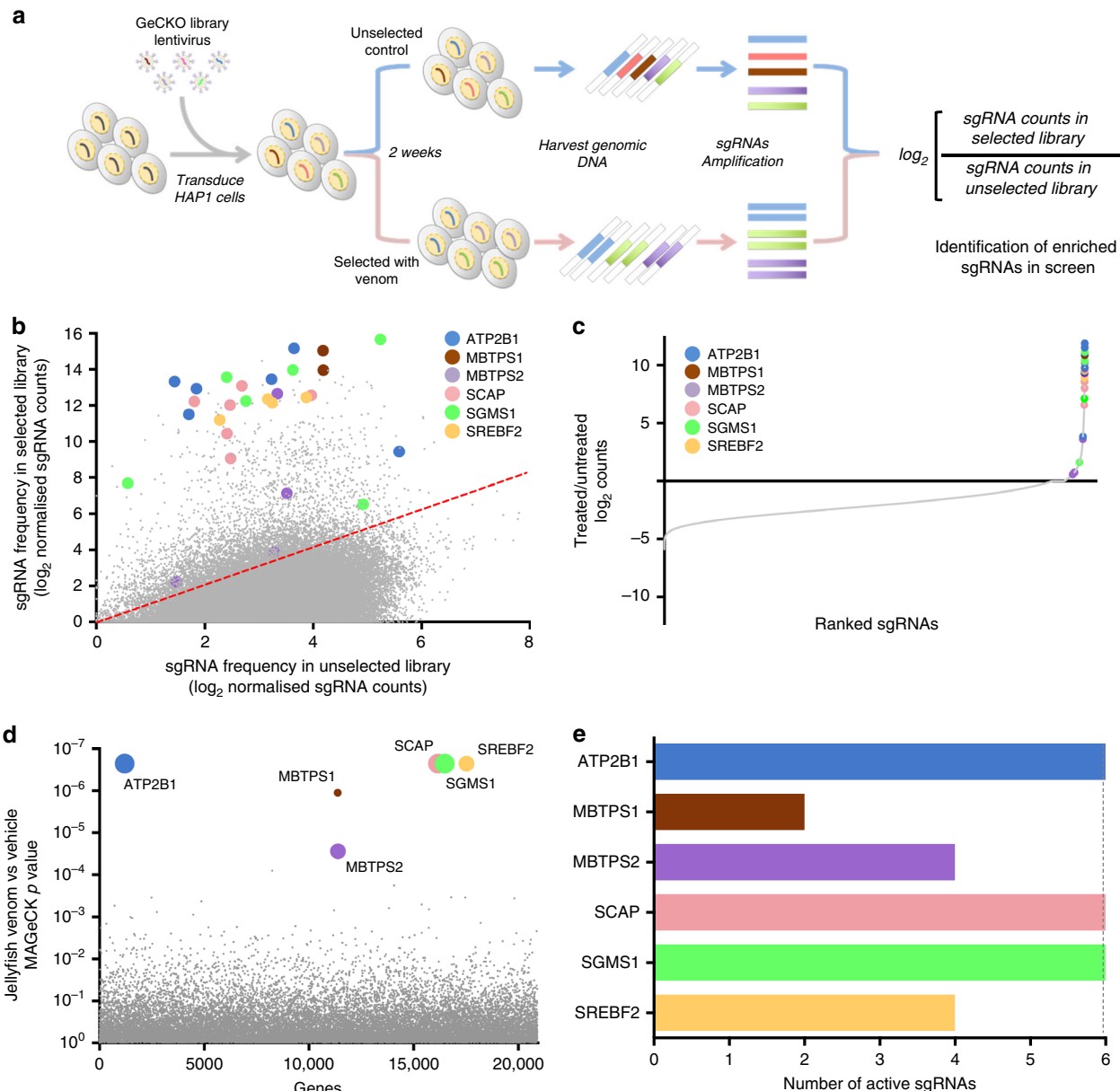

**Fig. 2** A CRISPR-Cas9 knockout screen identified genes required for jellyfish venom killing. **a** Schematic design of pooled CRISPR library screens to identify the genes required for jellyfish venom killing. **b** Primary jellyfish venom screening data showing enrichment of specific sgRNAs after the venom selection. The count for a sgRNA is defined as the number of reads that perfectly match the sgRNA target sequence. **c** sgRNAs were ranked by their differential abundance between the treated versus untreated populations. **d** Top candidate genes identified by comparing differential abundances of all sgRNAs between the treated versus untreated populations using MAGeCK. Bubble size of top hits is proportional to the number of active sgRNA per gene. **e** The number of active sgRNAs for each of the screen hits

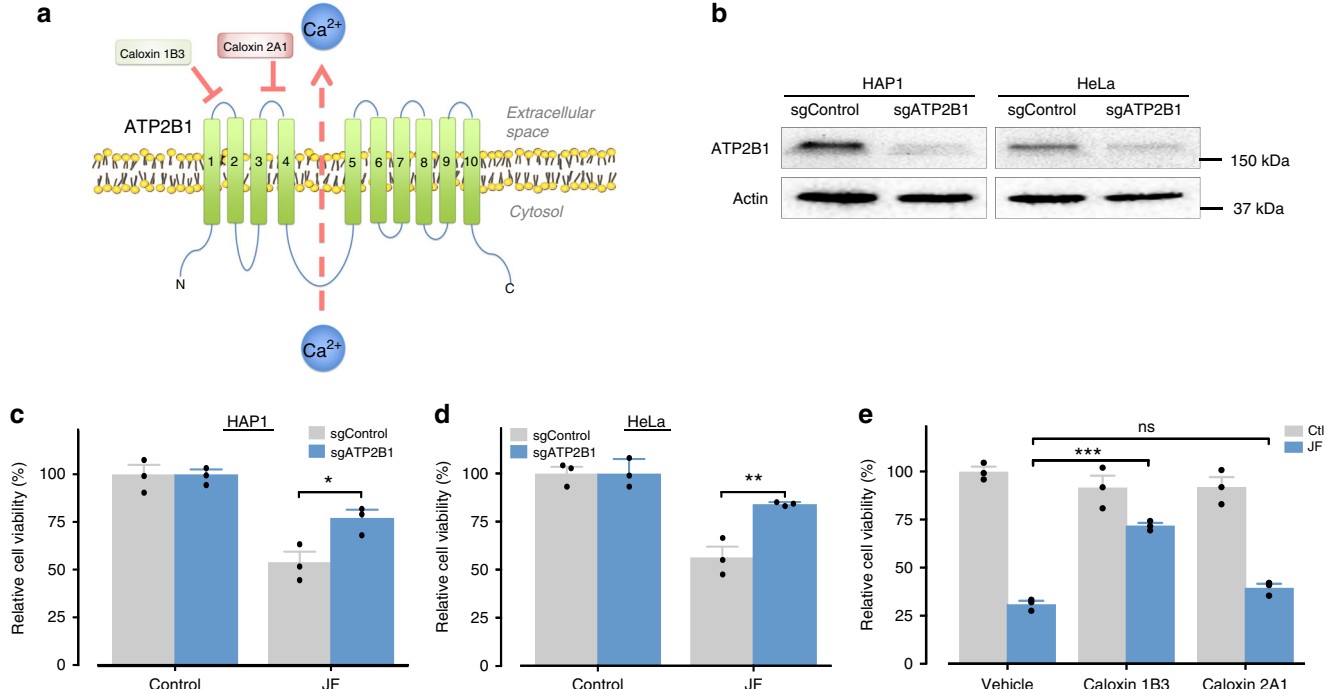

**Fig. 3** ATP2B1 is involved in jellyfish venom killing. **a** Schematic representation of ATP2B1. **b** Western blot validation of sgRNA-mediated depletion of ATP2B1 in HAP1 and HeLa cells. **c, d** Depletion of ATP2B1 conferring resistance to jellyfish venom in **c** HAP1 (0.75 µg/ml) and **d** HeLa cells (1 µg/ml). **e** Pre-treatment of caloxin 1B3 but not caloxin 2A1 reduces the jellyfish venom-induced cell death. All data represented as mean ± S.E.M ($n = 3$). **c, d** Student's $t$-test or **e** one-way ANOVA followed by Tukey's post hoc test, $*p < 0.05$; $**p < 0.01$; $***p < 0.001$; ns, not significant

Supplementary Data 2). To confirm a role for *ATP2B1*, we generated stable *ATP2B1* CRISPR knockout cell lines (Fig. 3b, Supplementary Fig. 2a). Importantly, depletion of ATP2B1 resulted in an increased resistance to the jellyfish venom in both HAP1 and HeLa cells (Fig. 3c, d, Supplementary Fig. 2b), confirming a role for this gene in venom cytotoxicity (Supplementary Fig. 2c–e). A role for ATP2B1 in venom cytotoxicity showed specificity for *C. fleckeri* venom, since disrupting ATP2B1 had no effect on cell death in response to venom from the sea nettle (*Chrysaora quinquecirrha*) a related jellyfish (Supplementary Fig. 2c), the pore-forming toxin streptolysin O (SLO) from *Streptococcus pyogenes* (Supplementary Fig. 2d), or α-hemolysin from *Staphylococcus aureus* (Supplementary Fig. 2e). ATP2B1 has multiple functional domains. Cells treated with caloxin 1B3, which targets the first extracellular loop of ATP2B1 were more resistant to jellyfish venom, while targeting the second extracellular loop with caloxin 2A1 had no effect (Fig. 3e). As ATP2B1 plays as essential role in the maintenance of intracellular $Ca^{2+}$ homeostasis[21], we tested if the box jellyfish venom kill cells via a calcium-dependent manner. Surprisingly venom triggered calcium influx independent of ATP2B1 (Supplementary Fig. 2f) and depletion of either external or internal calcium has no effect on venom cytotoxicity (Supplementary Fig. 2g, h). Thus, our data suggest that the box jellyfish venom works through ATP2B1 to induce cell death via a calcium-independent mechanism.

**Gene ontology (GO) terms and pathways analysis**. To provide better insight into the functional characteristics of the enriched screen hits, we performed GO analysis covering the following 3 categories: biological processes, cellular components and molecular functions (Fig. 4a–c; Supplementary Table 1). The highest enriched GO terms for biological processes included ER-nucleus signalling pathway, regulation of interleukin-18 production, sterol

metabolic process, membrane protein proteolysis, and glutamine family amino acid metabolic process (Fig. 4a). With respect to cellular component, the highest enriched GO terms included bounding membrane of organelle, Golgi apparatus part, an integral component of membrane, endoplasmic reticulum part, and mitochondrial intermembrane space (Fig. 4b). Whereas the highest enriched GO terms for molecular function included transforming growth factor beta binding, cAMP binding, protein homodimerisation activity, syntaxin binding, and cyclic nucleotide binding (Fig. 4c).

To further interrogate venom cytotoxic mechanisms, we performed pathway analysis based on the REACTOME database[22]. Our results identified 9 major pathways involved in jellyfish venom toxicity, including endosomal sorting complex required for transport (ESCRT), which is important for membrane budding into endosomes and lysosomes, subsequently leading to degradation. To confirm a role for endosomal function in venom-induced cell death, we blocked lysosome function with chloroquine, a drug known to neutralise the acidic lysosome environment[23]. Indeed, pre-treatment of chloroquine conferred resistance to the jellyfish venom cytotoxicity (Supplementary Fig. 3a). We further tested the potential therapeutic effect of chloroquine. This compound, however, only suppressed cell death when administered for 4 h before venom exposure (Supplementary Fig. 3b). Regulation of cholesterol biosynthesis by SREBP (SREBF) and sphingolipid de novo biosynthesis were also enriched in our venom resistance screen (Fig. 4d, e; Supplementary Table 1) and we next investigated a role for these processes in venom-induced cell death.

**Sphingomyelin is important for the venom-induced cell death.** To validate a role for sphingolipids in venom-induced cell death, we focused on *SGMS1*, a top-ranked hit (Fig. 2d) which encodes a key enzyme for sphingomyelin synthesis- sphingomyelin

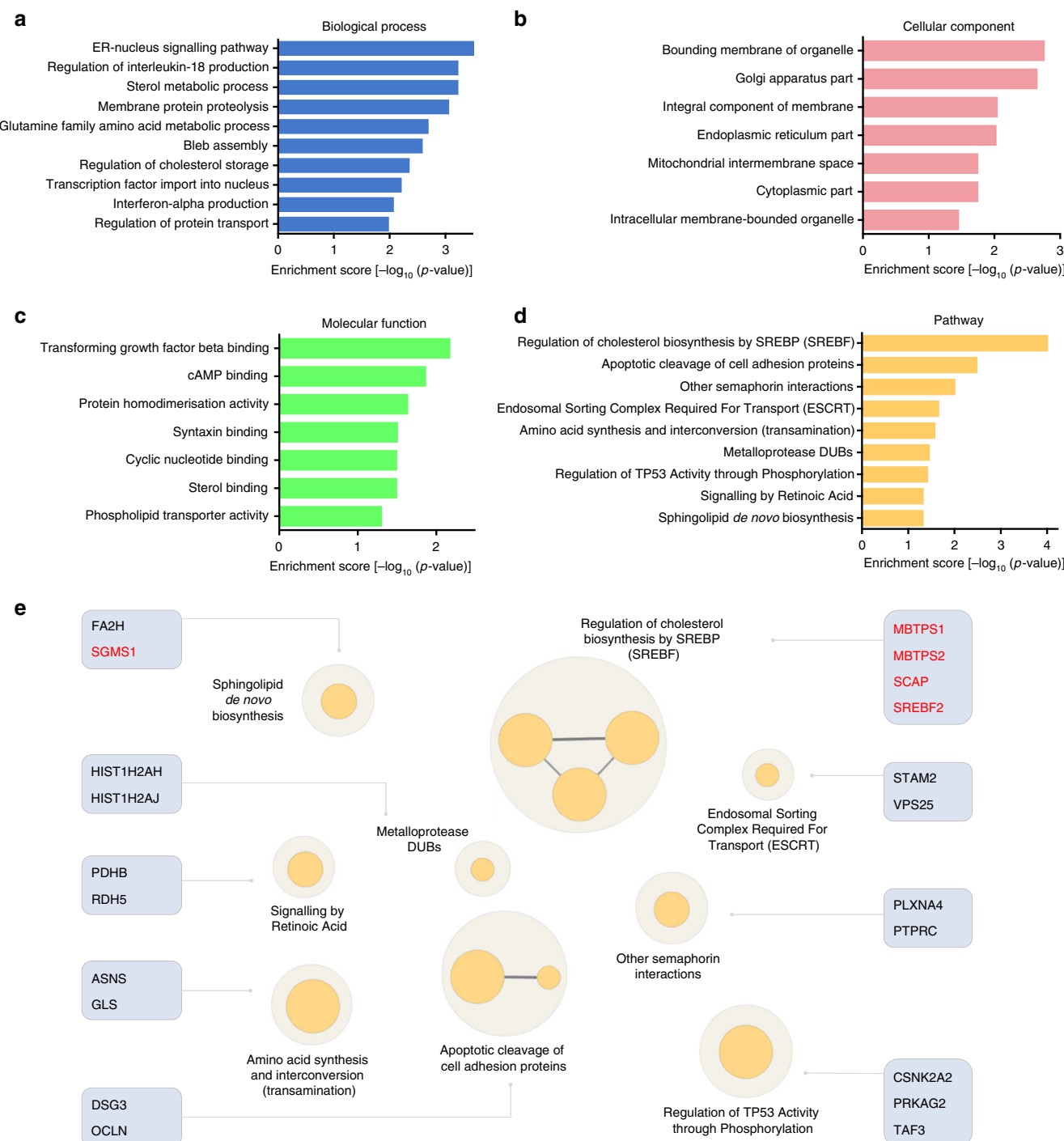

**Fig. 4** Enrichment analysis of gene ontology (GO) terms and pathways. **a–c** GO terms analysis of the significant screen hits according to **a** biological process, **b** cellular component and **c** molecular function. **d** Pathway analysis of screen hits based on the REACTOME database. **e** Enrichment network pathways were generated using the Cytoscape software[57] with ClueGo plug-in[58]. Groups are labelled according to the most significant pathways of the group and the related gene hits are showed. Genes that were top 10 hits are highlighted in red colour. The node size represents the term enrichment significance

synthase 1 (Fig. 5a). We generated *SGMS1*-deficient cells (Fig. 5b) and these cells were more resistant to the jellyfish venom treatment (Fig. 5c, d). We further investigated whether the depletion of cellular sphingomyelin can itself protect from the jellyfish venom. Indeed, cells pre-treated with sphingomyelinase (SMase), an enzyme that depletes sphingomyelin from cell membranes, were more resistant to the jellyfish venom, and became completely resistant to venom at the highest concentration used

(Fig. 5e). Together our data show that cellular sphingomyelin is critical for jellyfish venom cytotoxicity.

**Cholesterol is essential for the venom-induced cell death**. Our pathway analysis also highlighted regulation of cholesterol biosynthesis by SREBP as critical for venom cytotoxicity (Figs 3d, e and 6a) and four of 31 genes in this pathway were enriched in our

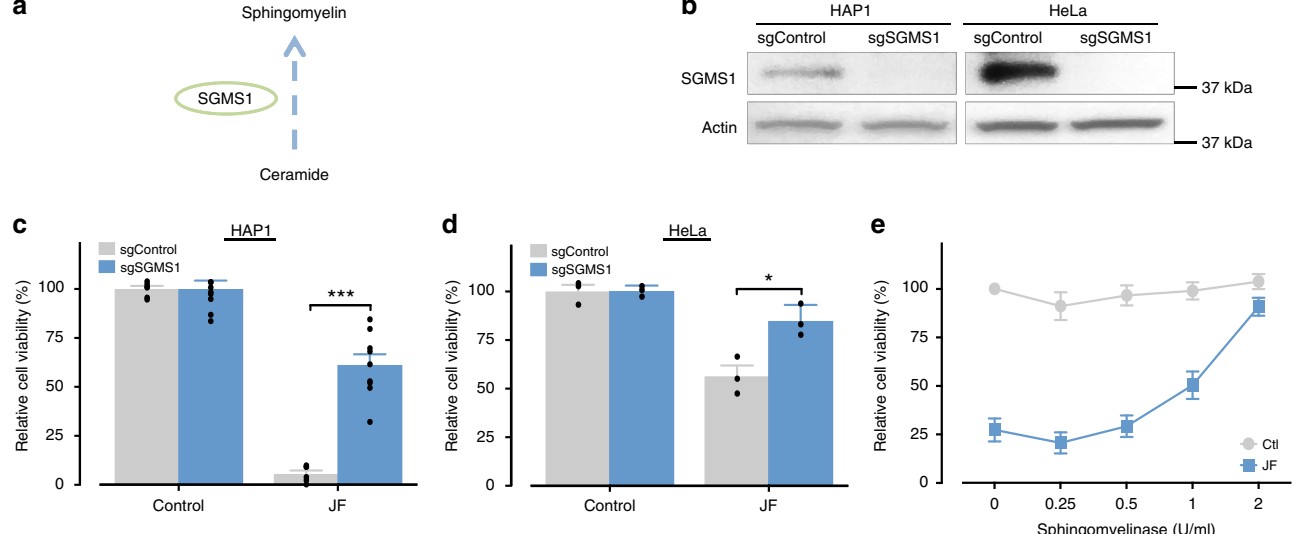

**Fig. 5** Sphingomyelin is required for jellyfish venom killing. **a** Schematic representation of SGMS1 in sphingomyelin biosynthesis. **b** Western blot validation of sgRNA-mediated depletion of SGMS1 in HAP1 and HeLa cells. **c**, **d** Depletion of SGMS1 conferring resistance to jellyfish venom in **c** HAP1 (2 μg/ml; $n = 6$) and **d** HeLa cells (1 μg/ml; $n = 3$). **e** The depletion of cellular sphingomyelin reduces jellyfish venom-induced cell death. HAP1 cells were pre-treated with the indicated concentration of sphingomyelinase (SMase) for 45 min, washed to remove SMase, and treated with jellyfish venom (1 μg/ml) for 24 h ($n = 3$). All data represented as mean ± S.E.M. Student's *t*-test, *$p < 0.05$; ***$p < 0.001$

screen (Supplementary Table 1). To elucidate this pathway for the jellyfish venom cytotoxicity, we generated several CRISPR-knockout lines. Importantly, depletion of key components in the SREBP pathway (*SCAP*, *MBTPS1* or *MBTPS2*) resulted in increased resistance to jellyfish venom confirming that these genes are required for venom cytotoxicity (Fig. 6b).

To independently determine the importance of cholesterol in jellyfish venom action, we used a pharmacological approach. Membrane cholesterol can be manipulated using methyl-β-cyclodextrin (MβCD) or 2-hydroxypropyl-β-cyclodextrins (HPβCD), compounds known to deplete cholesterol from cell membranes (Fig. 6c)[24–27]. As anticipated, cells treated with MβCD or HPβCD exhibited a concentration-dependent resistance to venom cytotoxicity (Fig. 6d, e) and a similarly these compounds also suppressed red blood cell haemolysis (Fig. 6f).

We next evaluated the therapeutic potential of these compounds. Importantly, these drugs suppressed cell death even when added up to 15 minutes after venom exposure (Fig. 7a, b). Jellyfish envenoming elicits significant pain and tissue necrosis[28] so we conducted in vivo experiments aimed at blocking these effects (Fig. 7c). HPβCD is a well-tolerated compound that has been used to treat Niemann-Pick Disease[27]. Injection of jellyfish venom into the mouse hind paw rapidly causes a dose-dependent spontaneous flinching indicative of pain (Supplementary Fig. 4a), and importantly HPβCD could suppress this response (Fig. 7d, Supplementary Fig. 4b). Jellyfish venom also had longer lasting effects, and 24 h after injection we observed tactile allodynia as assessed by von Frey mechanical touch assay (Supplementary Fig. 4c), this effect was slightly reduced by HPβCD (Fig. 7e). Importantly, we found the injection of jellyfish venom caused tissue necrosis at the site of envenoming (Fig. 7f–h, Supplementary Fig. 4d) and HPβCD had a potent ability to block tissue death (Fig. 7f–h, Supplementary Fig. 4d). Of note, jellyfish venom injection caused local swelling and this effect was not affected by HPβCD treatment (Supplementary Fig. 4e). Together, our unbiased whole-genome functional interrogation of the box jellyfish venom cell death pathway has highlighted multiple novel death mechanisms and guided the development of a new antidote for box jellyfish envenoming.

## Discussion

Aside from performing CPR, the classic treatment response for box jellyfish envenoming is to administer an antivenom in the emergency room[8], however, the efficacy of this treatment remains unclear. In this study, we used the power of genome-wide CRISPR screening to functionally isolate host components required for cell death after exposure to the lethal box jellyfish venom. Our unbiased screening revealed hundreds of candidate genes and host factors required for jellyfish venom cytotoxicity, many of which we have further validated both genetically and pharmacologically. From a systematic investigation of these data, we developed a rational box jellyfish venom antidote that can suppress tissue destruction and prevent pain associated with envenoming, the most common clinical manifestation of box jellyfish stings.

We report hundreds of human genes that may be targeted by box jellyfish venom components. For example, we identified the ATPase Plasma Membrane Ca²⁺ Transporter (*ATP2B1*) as a critical surface molecule required for venom-induced cytotoxicity. Our genetic and pharmacological evidence together argues that ATP2B1 interacts with a venom component, and this interaction is required for venom actions via a Ca²⁺-independent mechanism. Of note, while *ATP2B1* is necessary for cell death, a role for ATP2B1 in directly promoting pain remains to be shown. *ATP2B1* has been linked to regulation of hypertension in human populations[29], and smooth muscle-specific *ATP2B1* KO mice exhibit higher blood pressure[30]. With a deeper understanding of the mechanisms involved, venom components may be considered to modulate ATP2B1 as a next-generation anti-hypertensive strategy.

One critical insight provided by our genomic dissection of venom cytotoxicity is the role of cholesterol and sphingomyelin in the venom mechanism of action. Cholesterol and sphingomyelin are both major components of plasma membrane lipids[31,32] that serve as primary targets for numerous toxins. Previous studies demonstrated that the venom of jellyfish *Aurelia aurita* and *Cyanea capillata* have a preference to interact with cholesterol and sphingomyelin[33]. *C. fleckeri* venom contains pore-forming toxins that can cause haemolysis[6]. Pore-forming toxins can

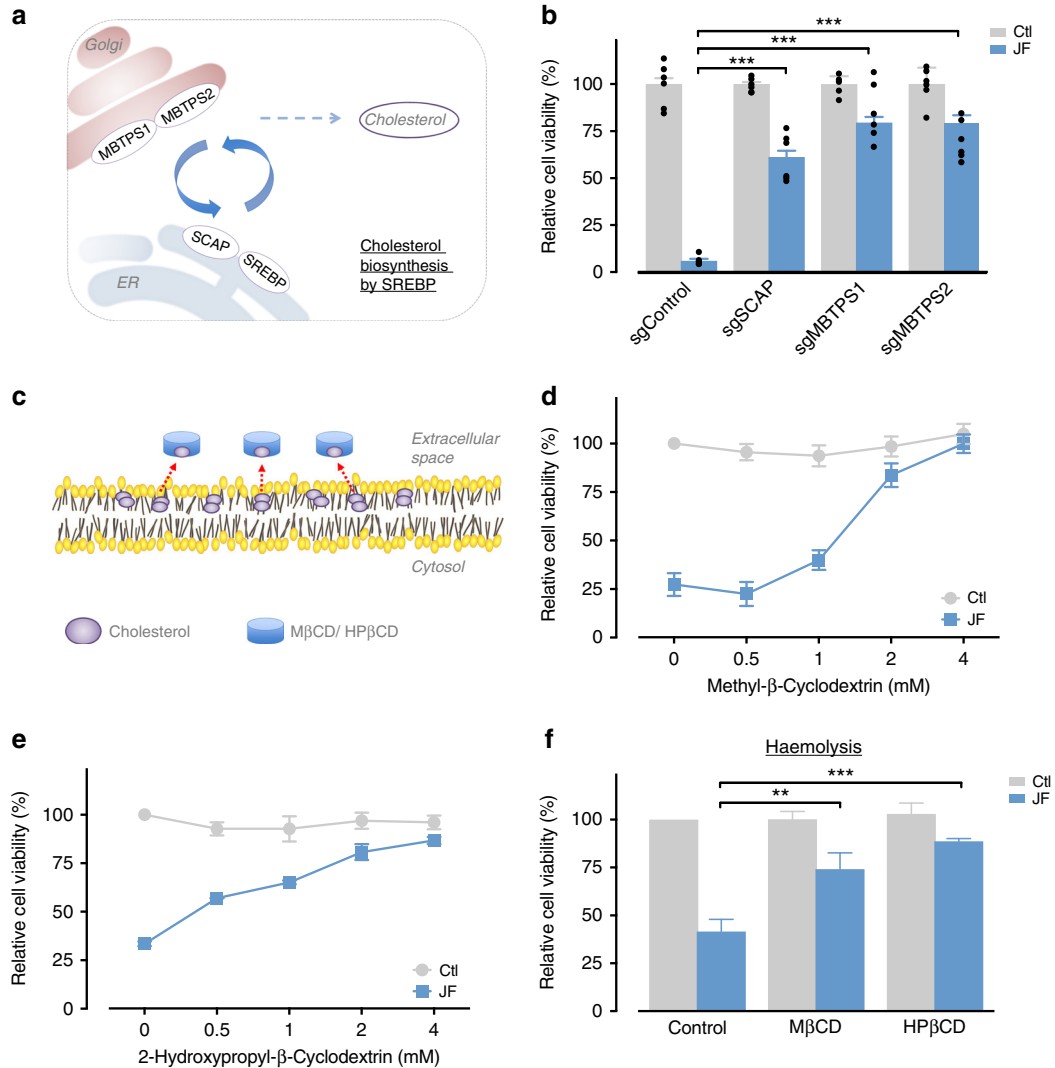

**Fig. 6** Cholesterol is required for jellyfish venom killing. **a** Schematic representation of the SREBP pathways in cholesterol biosynthesis. **b** Depletion of SCAP, MBTPS1 and MBTPS2 conferring resistance to jellyfish venom (2 μg/ml) in HAP1 cells ($n = 6$). **c** Schematic representation of the function of MβCD and HPβCD to remove cellular cholesterol. **d**, **e** The depletion of cellular cholesterol reduces jellyfish venom-induced cell death. HAP1 cells were pre-treated with the indicated concentration of **d** MβCD or **e** HPβCD for 45 min, washed to remove MβCD and HPβCD, and treated with jellyfish venom (1 μg/ml) for 24 h ($n = 3$). **f** Mouse erythrocytes were pre-treated with MβCD and HPβCD for 5 min and treated with jellyfish venom (2 μg/ml) for 1 h ($n = 3$). All data represented as mean ± S.E.M. One-way ANOVA followed by Tukey's post hoc test, *$p < 0.05$; **$p < 0.01$; ***$p < 0.001$

directly damage lipid membranes disrupting cell integrity, and also can provide an entry point for additional lethal venom components, eventually leading to cell death[34]. Our results showed that cholesterol or sphingomyelin depletion blocks box jellyfish venom cytotoxicity, and we establish MβCD and HPβCD as potential therapeutic antidote for box jellyfish envenoming. These substances have also been shown to block other bacterial pore-forming toxins[35,36]. Of note, depletion of cholesterol or sphingomyelin may affect the activity of jellyfish venom directly, or it may exert indirect effects through its role in membrane line tension and fluidity, and in the stabilisation of lipid rafts[37,38]. Consistent with these views, it has been demonstrated that ATP2B1 activity is functionally modulated by localisation to lipid rafts[39].

Alternatively, toxins can enter cells via the endocytic pathway, and subsequently trigger the lysosomal apoptotic pathway[40–43]. Our results highlighted gene sets related to endosomal sorting complex required for transport (ESCRT), suggesting the endocytic pathway is a possible mechanism for the uptake of the

jellyfish venom leading to lysosomal cell death. Moreover, blocking lysosomal function using chloroquine effectively protects cells from the jellyfish venom-mediated cell death and further refinement of venom constituents and potential endosomal targets and mechanisms of internalisation remain to be established.

Studies of bacterial toxin-induced apoptotic and necrotic mechanisms have been well documented[44]. However, the cytotoxic cell death mechanism triggered by box jellyfish venom is largely unknown. Other Jellyfish venoms cause cell death through osmolytic pathways[45] and the box jellyfish *Chriopsalmu quadrigatus* is reported to use apoptotic pathways[46]. Here we demonstrated that *C. fleckeri* venom-induced cell death occurs through multiple mechanisms. Our results showed that inhibition of MLKL, but not inhibition of caspases alone, reduced the box jellyfish venom-induced cytotoxicity, suggesting death occurs via necroptosis. Although inhibition of both pathways simultaneously was able to further prevent box jellyfish induced cytotoxicity potentially through convergent mechanisms[45]. Recently,

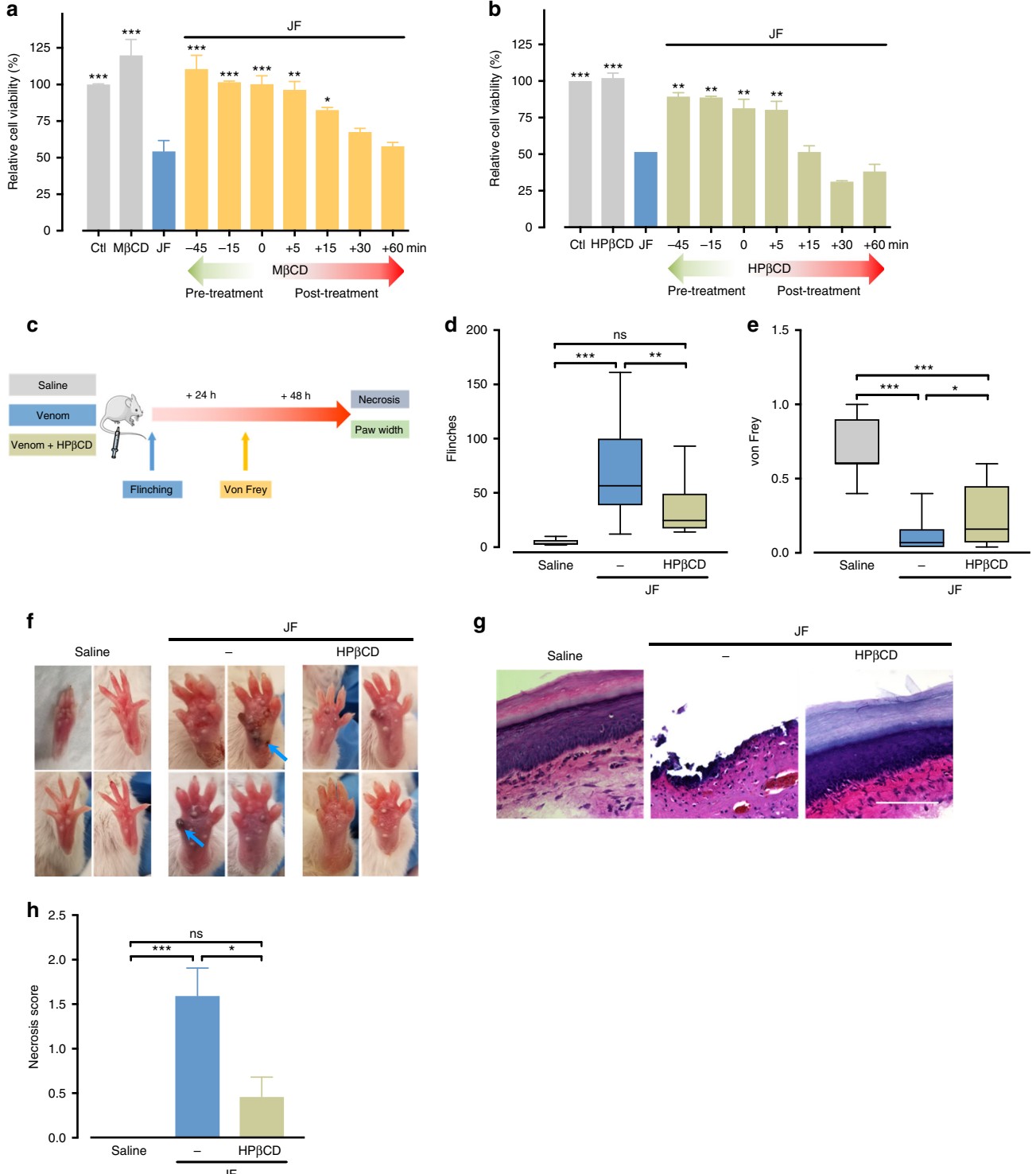

**Fig. 7** MβCD and HPβCD are potential box jellyfish venom antidotes. **a**, **b** Comparison of pre-treatment and post-treatment of **a** MβCD and **b** HPβCD in protection from jellyfish venom (0.75 μg/ml) in HAP1 cells ($n = 3$). Significance was assessed using one-way ANOVA followed by Tukey's post hoc test, **\****$p < 0.05$; **\*\****$p < 0.01$; **\*\*\****$p < 0.001$. **c** Schematic design of the in vivo study. This schematic figure was built using mouse templates available from the Servier Medical Art, which are licensed under a Creative Commons Attribution 3.0 Unported License; [https://smart.servier.com]. **d**, **e** Box plots showing pain response from **d** flinching behaviour and 50% paw withdrawal threshold **e** in von Frey test for mechanical hypersensitivity on mice treated with intraplantar injection of jellyfish venom alone or venom co-injected with HPβCD ($n = 7$–16; center line, median; box limits, upper and lower quartiles; whiskers, max to min). **f** HPβCD can suppress local tissue necrosis caused by injection of intraplantar jellyfish venom. **g** Haemotoxylin and Eosin staining of mouse footpad sections in control (Saline), venom treated (JF) and venom with HPβCD (JF & HPβCD) validating that venom treatment caused tissue necrosis. Scale bars are 100 μm. **h** quantification of venom necrosis in (**f**), Kruskall–Wallis followed by Dunn's post hoc test, **\****$p < 0.05$; **\*\****$p < 0.01$; **\*\*\****$p < 0.001$; ns, not significant. All data represented as mean ± S.E.M

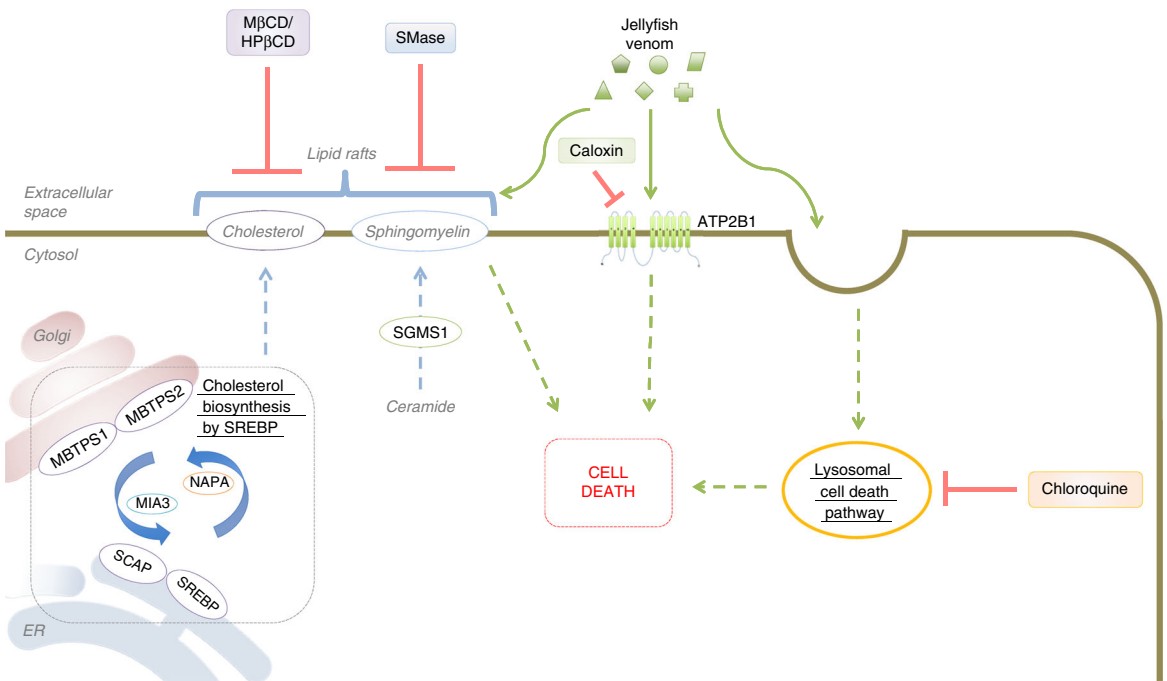

**Fig. 8** Schematic representation of the cellular pathways of top hit genes detected by the jellyfish venom screen

several toxins in *C. fleckeri* venom has been identified (Type I toxin: CfTX-1 and -2; Type II toxin: CfTX-A, -B and -Bt)[47],[48]. Comparative bioactivity assays revealed that the Type I toxin caused more potent cardiovascular collapse in anesthetised rats, whereas Type II toxins (including other box jellyfish species *Chiropsalmus quadrigatus*[49], *Carybdea rastoni*[50] and *Carybdea alata*[51]) associated with potent haemolytic activity and pore formation in mammalian erythrocytes[47]. To better understand box jellyfish envenoming, and help repurpose venom components as new medicines, a more in-depth analysis of the mechanisms of action for individual venom toxins using a similar GeCKO screen approach is warranted.

In summary, our unbiased screening approach has identified essential genes and cellular pathways involved in the jellyfish venom mechanisms of triggering cell death (Fig. 8) and lead to the identification of novel venom antidotes capable of suppressing pain and tissue destruction associated with envenoming. Further, we report likely interactions between jellyfish venom components and human factors, information that can provide an entry point to exploit jellyfish venom components as new medicines and propose that genomic dissection of venoms from diverse sources will accelerate drug discovery.

## Methods

**Materials**. Ac-DEVD-CHO was purchased from Selleck Chemicals. ATP2B1 antibody (#ab3528) was purchased from Abcam. Necrosulfonamide (NSA) and SGMS1 antibodies (#ABC732) were purchased from Merck Millipore. β-actin antibody (#4970) was purchased from Cell Signaling Technology, Inc. α-hemolysin, BAPTA-AM, Chloroquine, 2-hydroxypropyl-β-cyclodextrin (HPβCD), EDTA, methyl-β-cyclodextrin (MβCD), sea nettle (*Chrysaora quinquecirrha*) venom (SN), sphingomyelinase and streptolysin O (SLO) were purchased from Sigma-Aldrich. Caloxin 1B3 (TIPKWISIIQALRGGGSK-amide) and 2A1 peptide (VSNSNWPSFPSSGGG-amide) was prepared by custom synthesis from Mimotopes.

**C. fleckeri venom preparation**. *C. fleckeri* were collected from coastal waters near Darwin Harbour (Northern Territory, Australia). Nematocysts were extracted from excised jellyfish tentacles[52]. The box jellyfish venom was extracted from purified nematocysts, through a process of repeatedly exposing the nematocysts to 0.5 mm glass beads in a bead beater, followed by centrifugation at 3000 rpm for 1 min[53]. The suspended venom was aspirated, then lyophilised and stored at −80 °C.

**Cell culture**. Human HAP1 cells were generously provided by Dr. Thijn R. Brummelkamp[54]. HeLa cells were gift from Dr. Adam R. Cole, Garvan Institute. HAP1 and HeLa cells were cultured in Medium IMDM (Sigma-Aldrich) and DMEM (Sigma-Aldrich), respectively, containing 10% bovine calf serum (BCS; Hyclone Laboratories), 1x GlutaMAX, 100U/ml penicillin G and 100 g/ml streptomycin (Thermo Fisher Scientific) in a humidified atmosphere of 5% $CO_2$–95% air at 37 °C and were tested for the absence of mycoplasma contamination.

**Mice**. All mice in this study were male FVB/NJ mice aged 10–15 weeks obtained from the Animal Resource Centre, WA, Australia. All experiments were approved by the Animal Ethics Committee at the University of Sydney under protocol 1196. Mice were housed in a specific pathogen-free facility on a 12-hour light–dark cycle and standard chow, water and enrichment were provided ad libitum. All in vivo behavioural assays were performed blind to treatment by a single male investigator. Assignment to treatment groups was performed randomly by an investigator blind to the behaviour and health status of the animals.

**Cell viability assay**. Trypsinised cells ($3 \times 10^4$) were seeded in each well of a 96-well plate. After 24 h, various concentrations of jellyfish venom were added, and the cells were incubated for an additional 24 h. After incubation, the medium was aspirated from each well and add 150 μl of fresh medium containing a 0.002% solution of resazurin (Sigma-Aldrich) was added to the wells and incubated for 4 h at 37 °C. The absorbance was measured at 570 nm using a microplate spectrophotometer (FLUOstar Omega, BMG Labtech).

Lactate dehydrogenase (LDH) activity in supernatants of cells was assessed according to the protocol of the manufacturer (Thermo Fisher Scientific). Cell death was also determined by intracellular ATP levels using CellTiter-Glo Luminescent Cell Viability Assay (Promega) following the manufacturer's protocols.

**Haemolysis**. Mouse red blood cell (RBC) lysis was measured as described[47] with modification. Blood was collected by terminal cardiac puncture under isoflurane anaesthesia (2–3%) in MiniCollect EDTA coated tubes (Greiner-Bio-One). Collected blood was washed four times in sterile PBS and recovered by centrifugation at $500 \times g$ for 5 min at 4 °C. Diluted RBC (0.5% in PBS) were treated in triplicate in a 96-well plate and incubate for 1 h at 37 °C. Triplicates of 1% Triton X-100 and PBS alone were used as positive and negative control, respectively. Samples were centrifuged for 5 min at $500 \times g$ to pellet intact RBC, and supernatants were transferred to another 96-well plate. The absorbance of released haemoglobin was measured at 540 nm using a microplate spectrophotometer.

**Intracellular calcium measurement**. A Fluo-8 Calcium Flux Assay kit (Abcam) was used to measure intracellular calcium influx on a FLUOstar Omega microplate reader following the manufacturer's protocols. Briefly, trypsinised HAP1 cells ($3 \times 10^4$) were seeded in each well of a 96-well plate. After 24 h, plates were washed twice in $Ca^{2+}$-free HBSS supplemented with HEPES buffer (pH 7.2), and then the growth medium was replaced with 100 μl/well of the Fluo-8 dye solution. The plate was incubated at 37 °C for 30 min and then for another 30 min in the dark at room

temperature. The loaded cells were then placed in the measurement position in the microplate reader. Changes in fluorescence from the Fluo-8 dye quantify changes in intracellular $Ca^{2+}$ concentrations (excitation/emission 490/525 nm) after treatment with box jellyfish venom. $Ca^{2+}$ influx was measured up to 45 min.

**Lentivirus production**. To generate lentivirus, the human lentiCRISPRv2 plasmid library (Addgene 1000000048) was co-transfected with packaging plasmids pCAG-VSVG and psPAX2 (Addgene plasmids 35616 and 12260, respectively). Briefly, a T-75 flask of 80% confluent 293LTV cells (Cell Biolabs) was transfected in Opti-MEM (Thermo Fisher Scientific) using 8 µg of the lentiCRISPRv2 plasmid library, 4 µg pCAG-VSVG, 8 µg psPAX2, 2.5 µg pAdVantage (Promega), 30 µl of P3000 Reagent (Thermo Fisher Scientific), and 30 µl of Lipofectamine 3000 (Thermo Fisher Scientific). Cells were incubated overnight and then media was changed to DMEM (Sigma-Aldrich) with 10% BCS and 1x GlutaMAX (Thermo Fisher Scientific). After 48 h, viral supernatants were collected and centrifuged at 2000 rpm for 10 min to get rid of cell debris. The supernatant was filtered through a 0.45µm ultra-low protein binding filter (Merck Millipore). Aliquots were stored at −80 °C.

**Cell transduction using the GeCKO v2 library**. HAP1 cells were transduced with the GeCKO v2 library by spinfection. Briefly, $2 \times 10^6$ cells per well were plated into a 12-well plate in IMDM media supplemented with 10% BCS and 8 µg/ml polybrene (Sigma-Aldrich). A titrated virus was added in each well along with a no-transduction control. The plate was centrifuged at 2000 rpm for 1 h at 37 °C. After the spin, cells were incubated overnight and then enzymatically detached using TrypLE™ Express (Thermo Fisher Scientific). Cells were counted and each well was split into duplicate wells. One replicate treated with 1 µg/ml puromycin (Thermo Fisher Scientific) for 3 days. Percent transduction was determined as cell count from the replicate with puromycin divided by cell count from the replicate without puromycin multiplied by 100. The virus volume yielding a MOI (multiplicity of infection) approximately to 0.4 was used for large-scale screening.

**HAP1 jellyfish venom resistance screen**. $1 \times 10^8$ HAP1 cells were transduced as described above using 12-well plates with $2 \times 10^6$ cells per well. Puromycin was added to the cells 24 h post transduction and maintained for 7 days. Cells were pooled together into larger flasks after 3 days incubation of puromycin. On day 7, cells were split into treatment conditions in duplicate with a minimum of $2.5 \times 10^7$ cells per replicate. Two replicates were cultured in IMDM supplemented with 1 µg/ml jellyfish venom, and one replicate was culture in regular IMDM media. Replicates were either passaged or fresh media was added every 2–3 days. In parallel, untransduced HAP1 cells were treated with 1 µg/ml jellyfish venom to ensure the venom was cytotoxic in each case. Cells were harvested after 14 days of the treatment for genomic DNA analysis.

**Genomic DNA sequencing**. Genomic DNA (gDNA) extracted from harvested cells with a Blood & Cell Culture Midi kit (Qiagen) was used for PCR reactions as described previously[20]. Primers used to amplify lentiCRISPR v2 sgRNAs for the first PCR are: sense, 5′-AAT GGA CTA TCA TAT GCT TAC CGT AAC TTG AAA GTA TTT CG-3′ and antisense, 5′-TCT ACT ATT CTT TCC CCT GCA CTG TTG TGG GCG ATG TGC GCT CTG-3′.

A second PCR was performed to attach Illumina adaptors and to barcode samples. The second PCR was done in a 100 µl reaction volume with 5 µl of the first PCR product. Primers for the second PCR include both a variable length sequence to increase library complexity and an 6 bp barcode for multiplexing of different biological samples: sense, 5′-AAT GAT ACG GCG ACC ACC GAG ATC TAC ACT CTT TCC CTA CAC GAC GCT CTT CCG ATC T (1–9 bp variable length sequence) (6 bp barcode) tct tgt gga aag gac gaa aca ccg-3′ and antisense, 5′-CAA GCA GAA GAC GGC ATA CGA GAT AAG TAG AGG TGA CTG GAG TTC AGA CGT GTG CTC TTC CGA TCT tct act att ctt tcc cct gca ctg t-3′.

Amplification was carried out with 18 cycles for the first PCR and 24 cycles for the second PCR. PCR products from the second PCR were gel extracted, quantified, mixed and sequenced using a HiSeq 2500 (Illumina). The sgRNA sequences against specific genes were recovered after removal of the tag sequences using the Checkout [http://100bp.wordpress.com] and cutadapt (ver. 1.12).

Enrichment of sgRNAs and genes were analysed using MAGeCK[55] (Ver.0.5.6) by comparing read counts from cells after jellyfish venom selection with counts from matching unselected cell population to obtain a $p$-value. $P < 0.01$ was considered statistically significant.

**Gene validation**. To validate the candidate genes from screening, sgRNAs from the parent library were cloned into pLentiCRISPRv2 (Addgene plasmid 52961). The control sgRNA was used from the parent library. Lentiviruses were produced as described above and transduced HAP1 or HeLa cells were selected with 1 µg/ml puromycin 24 h post-infection. Two weeks later, puromycin was removed, and cells were allowed to recover for three additional days before analysis. Gene disruption efficiency was verified by western blot. The sequences of the sgRNAs used are in Table S4.

**Gene ontology (GO) and pathway enrichment analysis**. GO terms and REACTOME pathways enriched in the screen were analysed using the ConsensusPathDB[56]. The GO level 3–5 categories and a $p$-value cut-off of 0.05 were selected. The minimum overlap with the input list for pathway analysis was set at two proteins, with $p < 0.05$.

**Western blot analysis**. Cells were harvested in lysis buffer [50 mM Tris (pH 7.5), 150 mM NaCl, 1% NP40, 0.5% sodium deoxycholate, 1 mM EDTA, and 0.1% SDS] containing protease inhibitor cocktail (Sigma-Aldrich), and the protein concentrations were determined using the BCA Protein Assay (Thermo Fisher Scientific). The proteins (20 µg) were electrophoresed on 10% SDS-polyacrylamide gels, transferred to PVDF membranes (Amersham Bioscience), and incubated with specific primary antibodies (1:1000 for all primary antibodies) at 4 °C overnight. After washing, the membranes were incubated with horseradish peroxidase-conjugated secondary antibodies (1:5000; #31460 from Thermo Fisher Scientific) for 1 h and were then visualised with enhanced chemiluminescent substrate (Thermo Fisher Scientific). Uncropped images are shown in Supplementary Fig 5.

**Injections**. 250 ug of Jellyfish venom in 30 µl normal saline (0.9%, Pfizer) alone or in the presence of 20 ul HPβCD (50% W/V) was injected under isoflurane anaesthesia (2–3% mixed with 0.8 to 1 L $O_2$, IsoFlo Zoetis) into the mouse footpad. Rapid recovery from anaesthesia was achieved with high flow $O_2$.

**Spontaneous pain behaviour**. Following recovery, mice were placed into plexiglass boxes on a raised clear plexiglass table. Mouse behaviour was recorded from below using a video camera (Panasonic). The first five minutes of video were analysed for flinching behaviour (manually with the assistance of custom behaviour enumeration software) which was the dominant immediate pain behaviour.

**Von Frey mechanical allodynia assay**. Mice were habituated and tested with 10 probes of ascending filaments (Semmes Weinstein Filaments, North Coast Medical 0.04–2.0 g) on each hind paw in the sural territory. Mouse von Frey thresholds were established on 2 separate days for baseline readings and is defined as the stimulus eliciting reactions in 50% of probes. Mice were tested 1 day following injection of Jellyfish venom into the foot pad.

**Necrosis**. Necrosis was assessed 3 days following the injection of the venom. Photographs were taken of the mice hind paw and all photographs were scored blind. Necrosis was scored by 3 independent blinded investigators semi-quantitatively from pictures taken at euthanasia (either 3 days following injection or if an ethical endpoint was met). The score for each mouse is an average of the scores of each investigator. The scoring system used was:
    0) No evidence of necrosis
    1) Mild limited but limited to part of one digit or small portion of paw
    2) Mild affecting Multiple digits or affecting paw and digit or entire digit
    3) Moderate to severe affecting Multiple digits and paw and/or Autotomy
(any loss of a digit)

**Swelling**. Calipers were used twice daily to assess swelling, swelling scores are from just prior to euthanasia. The measurement was of the largest dorsoventral width.

**Histology**. Foot pads were dissected and fixed in paraformaldehyde 4% (Sigma) for 24 h then cryoprotected in 30% sucrose and flash frozen embedded in OCT (VWR) in Liquid nitrogen. The resultant tissue was then sectioned on a cryostat at 8–12 µm (Thermo Fisher). For Hoecst staining, tissue was washed in PBS, blocked in 2% BSA, 0.1% Triton X-100, washed and stained using Hoecst (Thermo Fisher 1:2000). Following this, tissue section was imaged at 40X using a Leica DM6000 upright microscope. For Haematoxylin and Eosin staining, PFA fixed frozen Sections were washed in water thoroughly to remove all OCT. The sections were immersed into Harris Haematoxylin for 30–40 s and washed in tap water until clear. The Haematoxylin was blued using Scott's Bluing solution and washed in tap water. The slides were immersed into ethanol then eosin. Finally, slides were dehydrated into 95% ethanol and 100% ethanol. The slides were dried then cleared into citrasol and cover-slipped using Xylene/citrasol and dried overnight and imaged using a DM6000 upright Leica microscope.

**Data analysis**. Statistical analysis performed was specified in figure legends. $p < 0.05$ was considered statistically significant.

## Data availability
The authors declare that all data supporting the findings of this study are available within the manuscript and its Supplementary Information files or are available from the authors upon reasonable request. A reporting summary for this study is available as a supplementary information file.

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

## Acknowledgements

M.T.L. was supported by a Cancer Institute New South Wales early career fellowship and G.G.N. is supported by an NHMRC career development fellowship II CDF1111940. This study was supported by Cancer Council NSW. We thank MacDonald Christie and Alexandra Sharland for sharing equipment and facilities. We also thank Laboratory Animal Services and the Design and prototyping workshop, and Sanaz Maleki, Sydney Microscopy and Microanalysis at the University of Sydney for supporting our experiments.

## Author contribution

M.T.L. designed the CRISPR knockout screen, designed and executed validation experiments, analysed and interpreted data; J.M. designed and performed the in vivo experiments with the assistance of J.B.L., L.O., T.M.K., Q.P.W., D.T.N. and D.H.; J.E.S. provided essential reagents; G.G.N. directed the study; M.T.L. and G.G.N. wrote the paper with comments from all the authors.

## Additional information

**Competing interests:** The authors declare no competing interests.

