## [Peer Review File · Nature Communications]

Reviewers' comments:

Reviewer #1 (Remarks to the Author):

The venom from *C. Fleckeri* causes substantial morbidity and even death in victims. This venom can cause tissue destruction and substantial killing of cells in tissue culture. The mechanism by which this venom kills cells in vitro is not known.

In this manuscript M-T Lau and colleagues report on a whole genome Crispr screen that they had conducted to identify cellular proteins and signaling pathways that are critical for *C. flicker* venom induced killing of cells in vitro. They also examined the role of one identified signaling pathway on venom induced morbidity in mice in vivo. The data shown are of substantial interest and mostly of good quality. I have some suggestions for additional experiments, which can all be done relatively easily (less than 3 months) and would, in my opinion, further increase the value of this paper.

1) the authors used zVAD-fmk to test if caspases are critical for venom induced killing of cells in vitro. They observed minor protection at 100 uM of this compound. I believe that this is not an on-target effect of zVAD-fmk. In fact at 20 uM or greater zVAD-fmk has been shown to kill cells in vitro in a manner that is unclear.

I suggest that the authors will also test the impact of the caspase inhibitor QVD-OPH, which as no reported off-target effects and no reported toxicity.

2) The authors should use Crispr to delete essential effectors of the intrinsic apoptotic pathway (BAX and BAK), the extrinsic apoptotic pathway (caspase-8; also need to delete MLKL to prevent cell killing due to aberrant necroptosis), necroptosis (MLKL) and proptosis (gasdermin D) to examine the individual roles of these cell death processes in venom induced cell killing.

If possible, the authors may even generate cell lines lacking two, three or all four processes to determine the overlapping roles of these pathways in venom induced killing of cells in vitro.

I believe such data would add great value to this paper.

3(There is a reported functional connection between the ESCRT pathway (identified by the authors as an important component of venom induced killing of cells in vitro) and pro-apoptotic BOK (F Llambi, Cell 2016). I suggest that the authors generate BOK deficient cell lines to examine this.

4) Figure 2B: needs molecular weight markers on the side

Also, why is the expression of ATP2B1 only reduced but not abrogated in the Crispr modified cell lines? I assume this is because the authors used polyclonal populations of targeted cells rather than clones. I suggest they select some clones that lack ATP2B1 and test the impact of the venom on these cells.

5) Figure 4b: : needs molecular weight markers on the side

6) Figure 6f: it would be good to add histological analysis of the tissues from the mice to show that HPbCD can indeed reduce venom induced tissue destruction

Reviewer #2 (Remarks to the Author):

General comments

While the work described in this manuscript is potentially interesting, the manuscript is poorly written and the conclusions are overstated. I am also concerned about the quality of the literature search that was performed in preparing the manuscript as much of the previous relevant literature has been overlooked or ignored. Two relatively old manuscripts (i.e. 1996 and 2003) are quoted to support the

study, whereas, a manuscript by Konstantakopoulos et al. J Pharmacol Toxicol Methods 59, 166, 2009 which also examines the effects of box jellyfish venom on cells is not mentioned.

Specific comments

Line 48. What do the authors mean by '...with significant skin contact'? Presumably they are referring to tentacle contact with the skin of human victims. This may result in discharge of the nematocysts and envenoming. If so, this should be clearly articulated. The current sentence is vague.

Line 51. What do the authors mean by '...jellyfish venom death pathway'? This appears to be a deliberate attempt to over dramatise the importance of this study. The authors are examining the viability of cell lines. 'Death' by *C. fleckeri* envenoming is almost certainly due to cardiovascular collapse. Whether this is due to an effect of the venom on cardiac or vascular tissue (or both) has not been elucidated although a number of studies have investigated this phenomenon.

Line 57. '...jellyfish-mediated venom activity'. What does this mean?

Line 61. The conclusion that the data presented in this manuscript will '...rapidly identify new medicines' is not supported by the results of the study.

Lines 74 and 207. What does 'classic treatment' mean? Surely antivenom is part of a recommended treatment strategy.

Line 76. '.....some actions of venom action'?

Replace 'envenomation' with 'envenoming' throughout.

Line 100. Replace 'dose-dependent' with 'concentration-dependent'.

The authors use the phrase 'cell death' throughout the manuscript. Is this what is being measured? Cell viability is used on the axis of the graphs. This seems to be a more accurate representation of the data being presented and discussed.

Lines 124 and 171. '...kill human cells'. Is this really what has been shown?

Lines 125 and 177. 'Jellyfish venom killing'!

Lines 202-203. '...of the box jellyfish venom death pathway'. This phrase is not appropriate and is an overly simplistic representation of the effects of envenoming and the data presented.

Lines 224-227. "...box jellyfish envenomation triggers a dramatic drop in blood pressure in vivo.....may be exploited as a next generation antihypertensive'. Where is the evidence in the current study (or reference to previous studies) that a decrease in BP occurs? In addition, the claim regarding next generation antihypertensive drugs is not supported by the current study and is a unsupported extrapolation of the effects of envenoming. i.e. many animal venoms/toxins cause profound effects on blood pressure given this is an effective method for subduing prey. This does not mean that they can be developed into pharmaceutical agents as per captopril.

Line 236 '...medicated'.

Lines 237-238. The statement re '...repurpose venom components as new medicines..' is not within the scope of this study.

Reviewer #3 (Remarks to the Author):

This study examined the molecular mechanisms by which box jellyfish (*Chironex fleckeri*) causes cell toxicity. The authors used genome-scale lenti-CRISPR mutagenesis to screen for host components required for cell death after venom exposure and identified peripheral membrane protein ATP2B1, a calcium transporting ATPase. They also identified a venom antidote that can suppress venom action. Overall, this is an interesting study and findings are novel. The genomic dissection of venom cytotoxicity is powerful. However, this study is also descriptive and lacks in-depth molecular mechanisms.

1. What are the venom toxins? Are these peptides?
2. The authors proposed an alternative pathway: "toxins can enter cells via the endocytic pathway, and subsequently trigger the lysosomal apoptotic pathway". However, the evidence provided is limited.
3. This study examined the venom-induced necrosis and pain in mice. However, it is unclear if pain and necrosis are mediated by distinct mechanisms. While ATP2B1 regulates necrosis, it may not cause pain. Toxins may cause pain via direct activation of ion channels.
4. The dose for venom injection (250 ug of Jellyfish venom in 30 μ l) is extremely high. Why do you need such a high dose for such a highly lethal venom? The dose for many pain inducers such as capsaicin and mustard oil is just 1 ug. A dose response curve is important. Also, a time course of spontaneous and evoked pain and development of necrosis will be informative.
5. In vivo data are limited. Are ATP2B1 KO mice available?

Other comments:

Abstract: "The box jellyfish *Chironex fleckeri* is the most venomous animal on the planet". Should this be restricted to "marine animal"?

Reviewer #4 (Remarks to the Author):

This manuscript describes the genome-wide interrogation of genes important for the cytolytic action of box jellyfish venom. It provides a comprehensive picture of the genes that contribute to the cytolytic activity thereby highlighting entirely novel mechanisms of toxin activity. The authors identify a specific plasma membrane calcium-transporting ATPase required to induce cell death upon intoxication. This gene has not before been linked to toxin-induced cell death and some more basic characterization beyond validation alone is warranted. The authors then focus on cholesterol and sphingomyelin as contributing factors. Finally they demonstrate in an in vivo model that cholesterol extraction by 2-hydroxypropyl- β -cyclodextrins reduced the pain and tissue necrosis associated with exposure to toxin. This well-written manuscript provides novel insights in the activity of a medically relevant toxin and goes from a genetic screen to a proof-of-principle that the identified genes could be targeted as potential therapeutic strategy.

Specific points:

1. The finding that ATP2B1 is required is really interesting given that this transporter has not been linked to cell death induced by other pore-forming toxins. Some basic characterization of how ATP2B1 functions would elevate the impact of the manuscript. In particular: Does calcium depletion from extracellular medium protect? Can the authors directly show calcium influx by fluorescence microscopy upon toxin exposure and how does this change upon ATP2B1 depletion? Do other well-characterized pore-forming toxins rely on ATP2B1?
2. The mode of cell death is only shown after using a mitochondrial activity assay. Since it is believed to be a pore-forming toxin, it would be informative to show cell death using an ATP depletion assay (e.g. CellTiter-Glo Luminescent Cell Viability Assay) and a membrane integrity assay (e.g. CytoTox-Glo™ Cytotoxicity Assay)
3. In the intro/discussion section, some more background on what is known of other box jellyfish toxins and how the mode of action uncovered by the screen compares to these and other classical pore forming toxins would be useful.
4. In the abstract the identified genes are called "Venom resistance genes". This incorrectly suggests that expression of these genes confers resistance.

Point-by-point reply to reviewers

Reviewer #1:

1) the authors used zVAD-fmk to test if caspases are critical for venom induced killing of cells in vitro... I suggest that the authors will also test the impact of the caspase inhibitor QVD-OPH, which as no reported off-target effects and no reported toxicity.

Response: As per the reviewer suggestion, in addition to zVAD-fmk we have now also tried both QVD-OPH and Ac-DEVD-CHO and in each case solely blocking caspase is not sufficient to block venom cytotoxicity, however we do find blocking caspase synergizes with blocking necroptosis. These data are presented in the revised manuscript (main Fig. 1c, d and Supplementary Fig. 1c, d).

2) The authors should use Crispr to delete essential effectors of the intrinsic apoptotic pathway (BAX and BAK), the extrinsic apoptotic pathway (caspase-8; also need to delete MLKL to prevent cell killing due to aberrant necroptosis), necroptosis (MLKL) and pyroptosis (gasdermin D) to examine the individual roles of these cell death processes in venom induced cell killing.

Response: this was a valuable suggestion, and we have now targeted BAK, BAX, BID, Caspase 8, gasdermin D, and MLKL. Importantly, MLKL was required for cell death in response to venom, and thus we conclude the death pathway involves necroptosis, however we also combined targeting of MLKL with caspase inhibition and observed additional effects on cell survival. These data are all included in the revised manuscript (see new Fig.1c, d, Supplementary Fig. 1e). We thank the reviewer for helping us improve our manuscript.

3) ESCRT pathway and pro-apoptotic BOK (F Llambi, Cell 2016). I suggest that the authors generate BOK deficient cell lines to examine this.

Response: As suggested we have generated BOK deficient cells and tested a role for BOK in venom cytotoxicity, however we did not observe any requirement for BOK in this process (see new Supplementary Fig. 1e).

3) Figure 2B: needs molecular weight markers on the side

Response: We have added molecular weight markers, please see new new Fig. 3b

4) authors used polyclonal populations of targeted cells rather than clones. I suggest they select some clones that lack ATP2B1 and test the impact of the venom on these cells.

Response: We have generated ATP2B1 clones and confirm a role for this gene in venom cytotoxicity, please see the data in new Supplementary Fig. 2a, b.

5) Figure 4b: : needs molecular weight markers on the side

Response: We have added molecular weight marker, please see new Fig. 5b.

6) *Figure 6f: it would be good to add histological analysis of the tissues from the mice to show that HPbCD can indeed reduce venom induced tissue destruction*

Response: We have included these data as requested in new Fig. 7g. Moreover, there is some degree of variability in venom mediated necrosis. This is captured quantitatively in Fig. 7h but we have also now included photos of multiple paws for control, venom, and venom plus drug. We feel together these data provide the reader with a complete understanding of the experimental results.

Reviewer #2:

1) *Two relatively old manuscripts (i.e. 1996 and 2003) are quoted to support the study, whereas, a manuscript by Konstantakopoulos et al. J Pharmacol Toxicol Methods 59, 166, 2009 which also examines the effects of box jellyfish venom on cells is not mentioned.*

Response: We have now added this additional reference. (see page 3 line 66).

2) *Line 48. What do the authors mean by '...with significant skin contact'? The current sentence is vague.*

Response: We have revised this sentence and removed “significant contact”. (see page 2 line 37).

3) *Line 51. What do the authors mean by '...jellyfish venom death pathway'?*

Response: We are referring to the jellyfish venom cell death pathway. We have edited the manuscript to address this concern. (see page 2 line 40).

4) *Line 57. '...jellyfish-mediated venom activity'. What does this mean?*

Response: This is referring to the cytotoxic activity of box jellyfish venom.

5) *Line 61. The conclusion that the data presented in this manuscript will '...rapidly identify new medicines' is not supported by the results of the study.*

Response: In this study we have used whole genome CRISPR pooled screening to rapidly identify a new medicine to treat box jellyfish envenoming. Thus, the conclusion is directly supported by the data.

5) *Lines 74 and 207. What does 'classic treatment' mean? Surely antivenom is part of a recommended treatment strategy.*

Response: Yes, classic treatment includes antivenom.

6) Line 76. '*.....some actions of venom action*'?

Response: revised to “some venom activities”. (see page 3 line 67).

7) Replace '*envenomation*' with '*envenoming*' throughout.

Response: Both are used in publications, but we have no preference and have changed to “envenoming” as requested.

8) Line 100. Replace '*dose-dependent*' with '*concentration-dependent*'.

Response: This has been changed as requested. (see page 4 line 90).

9) '*cell death*' vs .. *Cell viability*?

Response: Life and death are two sides of the same coin. As we evaluate viability we also gain insight into cell death.

10) Lines 124 and 171. '*...kill human cells*'. *Is this really what has been shown?*

Response: Yes.

11) Lines 125 and 177. '*Jellyfish venom killing*'!

Response: This has been addressed.

12) Lines 202-203. '*....of the box jellyfish venom death pathway*'. *This phrase is not appropriate and is an overly simplistic representation of the effects of envenoming and the data presented.*

Response: We have revised this and other statements to clarify that we are referring to cell death.

13) Lines 224-227. "*...box jellyfish envenomation triggers a dramatic drop in blood pressure in vivo.....may be exploited as a next generation antihypertensive*'. *Where is the evidence in the current study (or reference to previous studies) that a decrease in BP occurs? In addition, the claim regarding next generation antihypertensive drugs is not supported by the current study and is a unsupported extrapolation of the effects of envenoming. i.e. many animal venoms/toxins cause profound effects on blood pressure given this is an effective method for subduing prey. This does not mean that they can be developed into pharmaceutical agents as per captopril.*

Response: We have removed mention of venom triggering decreased blood pressure and have edited this paragraph to highlight that using jellyfish venom to treat blood pressure may be a possible but not certain outcome of this work. (see page 8 line 229-238).

14) Line 236 *'..medicated'*.

Response: This has been corrected.

15) Lines 237-238. *The statement re '....repurpose venom components as new medicines..' is not within the scope of this study.*

Response: We highlight that our work here provides the initial entry point for future work developing jellyfish venom components as potential medicines. Since we have identified direct or indirect machinery used by jellyfish venom, some of these venom constituents could become lead compounds if these targets are found worthy of consideration as therapeutic targets. Our data supports this potential outcome.

Reviewer #3:

1). *What are the venom toxins? Are these peptides?*

Response: The components in the box jellyfish venom is still largely unknown. Our results shown that boiled venom has no cytotoxic effect on HAP1 cells (data not shown) suggesting that these are proteins / peptides. We have added more details about the jellyfish venom in the revised manuscript (see page 3 line 59-61).

2). *The authors proposed an alternative pathway: "toxins can enter cells via the endocytic pathway, and subsequently trigger the lysosomal apoptotic pathway". However, the evidence provided is limited.*

Response: We agree with the reviewer's statement. At this stage, we do not have direct evidence showing jellyfish toxins/ venom enter cells via the endocytic pathway. We have now clarified this in the revised manuscript (see page 9 line 262-263).

3). *This study examined the venom-induced necrosis and pain in mice. However, it is unclear if pain and necrosis are mediated by distinct mechanisms. While ATP2B1 regulates necrosis, it may not cause pain. Toxins may cause pain via direct activation of ion channels.*

Response: We agree with the reviewers statement. Pain may arise downstream of venom necrotic activities, or venom may directly cause pain independent of necrosis, or both. Our data with the drug (HPbCD, new Fig 7) suggests that if we block necrosis we substantially suppress pain. At this stage we do not have direct evidence linking ATP2B1 to pain. We have now clarified this in the revised manuscript (see page 8 line 234-235).

4). *The dose for venom injection (250 ug of Jellyfish venom in 30 ?l) is extremely high. Why do you need such a high dose for such a highly lethal venom? The dose for many pain inducers such as capsaicin and mustard oil is just 1 ug. A dose response curve is important. Also, a time course of spontaneous and evoked pain and development of necrosis will be informative.*

Response: We agree with the reviewer this is an important issue. We now provide a dose response curve for spontaneous and evoked pain (Supplementary Fig. 4a, c), and the results shown that 250ug of jellyfish venom induces the pain consistently. The time course of spontaneous and evoked pain and development of necrosis data are shown in Supplementary Fig. 4b,d. As for the elevated dose compared to capsaicin, we do not have a definitive response. It is possible that since the venom causes pain in many marine or terrestrial species, the venom component is not as selective as capsaicin. Alternatively, it could be that pain is secondary to necrotic tissue damage. We are currently trying to purify the pain and necrotic components of this venom, however this project will take years and is beyond the scope of this study.

5). *In vivo data are limited. Are ATP2B1 KO mice available?*

Response: *Atp2B1* KO mice are embryonic lethal and die during preimplantation.

6). *Abstract: “The box jellyfish Chironex fleckeri is the most venomous animal on the planet”. Should this be restricted to “marine animal”?*

Response: The box jellyfish is often referred to as the most venomous animal on the planet, however we acknowledge that this may be debateable, and we have revised this statement to “one of the most venomous ...”. (see line 36 and 55).

Reviewer #4:

1.) *Does calcium depletion from extracellular medium protect?*

Response: This was an important issue, we have performed experiments removing intracellular or extra calcium and neither perturbation was required for venom-induced cell death. These data are found in Supplemental Fig 2g, h.

2) *Can the authors directly show calcium influx by fluorescence microscopy upon toxin exposure and how does this change upon ATP2B1 depletion?*

Response: Yes, venom induces calcium flux, however this is independent of ATP2B1, suggesting that the venom works through ATP2B1 via a calcium-independent mechanism (Supplemental Fig. 2f).

3) *Do other well-characterized pore-forming toxins rely on ATP2B1?*

Response: We tested if ATP2B1 is a host factor for other jellyfish venom² (sea nettle) or pore-forming toxins (such as streptolysin O and α -hemolysin), our results shown that ATP2B1 is only required for box jellyfish venom. Moreover, a previous study for α -hemolysin toxicity did not reveal ATP2B1 from the genome-wide CRISPR screen³. These data can be found in Supplemental Fig. 2c-e.

2. *The mode of cell death is only shown after using a mitochondrial activity assay. Since it is believed to be a pore-forming toxin, it would be informative to show cell death using an ATP depletion assay (e.g. CellTiter-Glo Luminescent Cell Viability Assay) and a membrane integrity assay (e.g. CytoTox-Glo™ Cytotoxicity Assay)*

Response: We have now included data from an ATP depletion assay and LDH cytotoxicity assay (a membrane integrity assay) to further confirm that the box jellyfish venom induced cell death *in vitro* (Supplementary Fig. 1a and b).

3. *In the intro/discussion section, some more background on what is know of other box jellyfish toxins and how the mode of action uncovered by the screen compares to these and other classical pore forming toxins would be useful.*

Response: We have included more background on what is known for box jellyfish venoms, we thank the reviewer for this suggestion and feel this has improved the manuscript. (line 59-61; line 245-251; line 265-278).

4. *In the abstract the identified genes are called “Venom resistance genes”. This incorrectly suggests that expression of these genes confers resistance.*

Respond: We agree and have revised the abstract to address this issue in the abstract and main text.

References

- 1 Ziegler, A. B. *et al.* Cell-Autonomous Control of Neuronal Dendrite Expansion via the Fatty Acid Synthesis Regulator SREBP. *Cell Rep* **21**, 3346-3353, doi:10.1016/j.celrep.2017.11.069 (2017).
- 2 Ponce, D., Brinkman, D. L., Potriquet, J. & Mulvenna, J. Tentacle Transcriptome and Venom Proteome of the Pacific Sea Nettle, *Chrysaora fuscescens* (Cnidaria: Scyphozoa). *Toxins (Basel)* **8**, 102, doi:10.3390/toxins8040102 (2016).
- 3 Virreira Winter, S., Zychlinsky, A. & Bardoel, B. W. Genome-wide CRISPR screen reveals novel host factors required for *Staphylococcus aureus* alpha-hemolysin-mediated toxicity. *Sci Rep* **6**, 24242, doi:10.1038/srep24242 (2016).

REVIEWERS' COMMENTS:

Reviewer #1 (Remarks to the Author):

The authors have performed all experiments that I have asked them to perform. I have no further requests.

Reviewer #2 (Remarks to the Author):

I am satisfied that the authors have adequately addressed my concerns

Reviewer #3 (Remarks to the Author):

The authors are responsive by including new data showing dose-dependent effects of toxin on spontaneous pain as well as the time course of pain.

Reviewer #4 (Remarks to the Author):

The authors have satisfactorily addressed all my concerns and I support publication. The already strong manuscript has been significantly improved by the inclusion of more mechanistic data defining better the mode of cell death by the toxin as well as the surprising finding that ATP2B1 seems to act in a calcium independent fashion.

Several minor suggestions that do not require new experiments:

1. If available, please include Western Blots showing depletion of BAK, BAX, BID, Caspase 8, gasdermin D, and MLKL
2. necrosulfonamide is misspelled on line 95
3. Please always mention the concentration of venom used (e.g. in Fig. 1c, d)

REVIEWERS' COMMENTS:

Reviewer #1 (Remarks to the Author):

The authors have performed all experiments that I have asked them to perform. I have no further requests.

Reviewer #2 (Remarks to the Author):

I am satisfied that the authors have adequately addressed my concerns

Reviewer #3 (Remarks to the Author):

The authors are responsive by including new data showing dose-dependent effects of toxin on spontaneous pain as well as the time course of pain.

Reviewer #4 (Remarks to the Author):

The authors have satisfactory addresses all my concerns and I support publication. The already strong manuscript has been significantly improved by the inclusion of more mechanistic data defining better the mode of cell death by the toxin as well as the surprising finding that ATP2B1 seems to act in a calcium independent fashion.

Several minor suggestions that do not require new experiments:

1. If available, please include Western Blots showing depletion of BAK, BAX, BID, Caspase 8, gasdermin D, and MLKL

RESPONSE: We did not have the available budget to purchase antibodies for all genes targeted, and as such we do not have western blots showing depletion for these factors. If this is considered essential by the reviewer, we can purchase these antibodies and perform the requested blots.

2. necrosulfonamide is misspelled on line 95

RESPONSE: We have corrected the spelling for “necrosulfonamide”.

3. Please always mention the concentration of venom used (e.g. in Fig. 1c, d)

RESPONSE: We have included the concentration of venom used in the figure legends.